# Effects of Water and Nitrogen Management on Water Productivity, Nitrogen Use Efficiency and Leaching Loss in Rice Paddies

**Kaiwen Chen** [ID]**, Shuang'en Yu \*, Tao Ma \***[ID]**, Jihui Ding, Pingru He** [ID]**, Yan Dai and Guangquan Zeng**

College of Agricultural Sciences and Engineering, Hohai University, Nanjing 210098, China;
kwchen@foxmail.com (K.C.); dingjihui@hhu.edu.cn (J.D.); hepingru68@163.com (P.H.); daiyan@hhu.edu.cn (Y.D.);
zengqhalo@outlook.com (G.Z.)
* Correspondence: seyu@hhu.edu.cn (S.Y.); matao@hhu.edu.cn (T.M.)

**Abstract:** Effective water and nitrogen (N) management strategies are critical for sustainable agricultural development. Lysimeter experiments with two deep percolation rates (low percolation and high percolation, i.e., LP and HP: 3 mm d$^{-1}$ and 5 mm d$^{-1}$) and five N application levels (N0~N4: 0, 60, 135, 210 and 285 kg N ha$^{-1}$) were conducted to investigate the effects of controlled drainage on water productivity (*WP*) and N use efficiency (*NUE*) in water-saving irrigated paddy fields. The results demonstrated that $NH_4^+$-N and $NO_3^-$-N were the major components of total nitrogen (TN) in ponded water and leachate, accounting for more than 77.1% and 83.6% of TN, respectively. The risk of N leaching loss increased significantly under treatment of high percolation rates or high N application levels. High percolation loss required greater irrigation input, thus reducing *WP*. In addition, N uptake increased with increasing N application, but fertilization applied in excess of crop demand had a negative effect on grain yield. *NUE* was affected by the amount of N applied and increased with decreasing N levels. Water and N application levels had a significant effect on N uptake of rice, but their interaction on N uptake or *NUE* was not significant. For the LP and HP regimes, the highest N uptake and *WP* were obtained with N application levels of 285 kg ha$^{-1}$ and 210 kg ha$^{-1}$, respectively. Our overall results suggested that the combination of controlled drainage and water-saving irrigation was a feasible mitigation strategy to reduce N losses through subdrainage percolation and to provide more nutrients available for rice to improve *NUE*, thus reducing diffuse agricultural pollution. Long-term field trials are necessary to validate the lysimeter results.

**Keywords:** water-saving irrigation; percolation; nitrogen concentration; leaching loss; rice yield; nitrogen uptake; controlled drainage

## 1. Introduction

Rice (*Oryza sativa* L.) is one of the main cereal crops in the world, feeding about half of the global population [1]. In China, rice accounts for 35% of cereal production [2], which alone consumes about 50% of the freshwater resources [3] and more than 11 million tons of nitrogen (N) fertilizers [4]. To ensure the high and stable yield of rice, the traditional continuous flooding irrigation method with a large dose of N-fertilizer was adopted in most regions. It consumed 50~300 cm of field water input during rice growth [5]. However, due to urban expansion, population growth and increasing extreme climate events, the security of agricultural water supply is under threat, which has brought challenges to the sustainable development of irrigated agriculture [6,7]. Moreover, excessive application of N fertilizer and improper water management causes excessive losses of water and nutrients from runoff and deep percolation (DP) in rice paddies [8,9], which has drawn attention to ecological and environmental problems such as soil acidification and accelerated eutrophication of lakes [10–12].

Due to water resource shortage, many rice water-saving irrigation strategies have been introduced [13–15], including alternate wetting and drying (AWD) irrigation [16], which dynamically regulates the wetting or drying state of the paddy field during rice growth [5] to improve water productivity (*WP*). In southern China, the rice-growing season is consistent with the summer rainy season with an average annual precipitation of more than 1000 mm. Considering the potential contribution of rainfall to irrigation, reasonable rainwater management with controlled drainage is an alternative option to improve *WP* in irrigated agriculture, which is also beneficial to reducing the rate and volume of surface runoff, stormwater mitigation and enhancing denitrification in soil [17]. Combining the advantages of AWD, the controlled irrigation and drainage (CID) practice has been proposed for the humid climate characteristics of southern China [18,19], which maintains a higher water depth and captures more drainage in paddy fields during rain events. Under CID, however, the paddy soil is frequently subjected to more severe periodic dry/wet cycles than AWD. More ponding rainwater and longer ponding time after a period of soil moisture deficit may lead to a decline in yield due to the poor aeration conditions of rice roots. In addition, DP is increased with the high water head of the paddy surface, and the corresponding N leaching loss poses a threat to the groundwater environment [20]. When controlled drainage is coupled with rice water-saving irrigation, the alternation of soil water deficit and flooding is intensified, which changes the microbial processes such as nitrification and denitrification [21], thus bringing more uncertainty to rice N uptake and growth.

Besides water, N is another production input for rice, and the pursuit of high N use efficiency (*NUE*) is necessary for high yield and environmental protection. In China, the apparent N recovery efficiency (*RE*) of rice during the past 10 years was only 39% [22], and the partial factor productivity of N (*PFP*) has dropped from 55.0 kg ha$^{-1}$ to 20.0 kg ha$^{-1}$ from 1977 to 2005 [23] due to improper use of N-fertilizers. Excessive N application along with conventional water management practices can easily lead to serious N loss from rice paddies and eutrophication of surface and underground water. Han et al. [24] reported that rice plant uptake and nitrification-denitrification loss accounted for 68.0~75.0% and 5.1~9.3% of N output, respectively, under AWD-based strategies. In addition, *NUE* is a comprehensive trait influenced by the interaction of environmental factors and biochemical pathways [25]. Some studies have indicated that synergistic interaction existed between water and N to improve *NUE* and crop yield [26,27]. However, there were also studies suggesting that increased ammonia volatilization and leaching loss of N by water-saving irrigation ultimately reduced cumulative plant N uptake and *NUE* [28,29]. In addition, high N-fertilizer application may aggravate stress and thus have a negative impact on grain yields in the case of limited soil moisture [30]. It is hypothesized that the combined application of water-saving irrigation and controlled drainage strategies under the CID regime may affect the *WP* and *NUE* of rice at a certain N application rate. However, the relevant evidence is scarce.

The present study was to identify the N leaching loss, N uptake and *NUE*, and compare the grain yield and *WP* of water-saving irrigated rice fields after the introduction of controlled drainage. Two DP rates under the CID regime (LP and HP, i.e., low percolation and high percolation, representing 3 mm d$^{-1}$ and 5 mm d$^{-1}$ subsurface percolation rates, respectively) and five N-fertilizer rates (N0~N4, representing 0, 60, 135, 210, and 285 kg N ha$^{-1}$ application, respectively) with straw return were set, and rice yield and N content associated with field environment were measured, including ammonium-N ($NH_4^+$-N) and nitrate-N ($NO_3^-$-N) and total nitrogen (TN) in water samples, and total N accumulation by rice above-ground organs. The information obtained from this study will scientifically guide water and N management and sustainable agricultural development of humid paddy fields.

## 2. Materials and Methods

### 2.1. Experimental Site Description

The experiments were conducted at the Jiangning Water-saving Park, Jiangsu Province, China (31°95′ N, 118°83′ E, altitude of 15.0 m) during the rice growth season (June to October) in 2019, which was laid out in 32 lysimeter plots, with length, width and depth of 2.0 m, 2.5 m and 2.0 m, respectively. Figure 1 shows a typical cross-section of lysimeter soils in the vertical direction. It is evident that the surface layer (0~20 cm) maintains water after irrigation, experiencing a periodic cycle of alternating dry and wet, where the rice roots are most densely distributed. The plough pan (20~40 cm) with very low permeability, formed by puddling or plowing action over the years of rice cultivation, prevents water from infiltrating deeper into the soil profile and reduces the water supply to the lower layers, and accordingly, it reduces the downward migration of solute. It should be noted that the cultivation of rice-wheat rotations on the lysimeter plots has been maintained for several years, and the soil properties have been tested and listed in Table 1. In addition, 40 cm thick coarse sand was laid at the bottom of each concrete lysimeter, where a water-permeable pipe with a valve was embedded to collect the DP. To measure the dynamics of the groundwater level, PVC (Poly Vinyl Chloride) pipes (inner diameter 42 mm) were drilled and embedded in lysimeters to perform as a field water level gauge. The well-equipped lysimeter experiment was a good approximation of the controlled drainage conditions in the field environment. This site has a subtropical and humid climate, with an average annual temperature of 15.3 °C, an average annual rainfall of 1051 mm, and a sunshine duration of 2212.8 h. Daily reference evapotranspiration, air temperature and precipitation are depicted in Figure 2, and six of these rainfall events reached the heavy rain level (>25 mm per day).

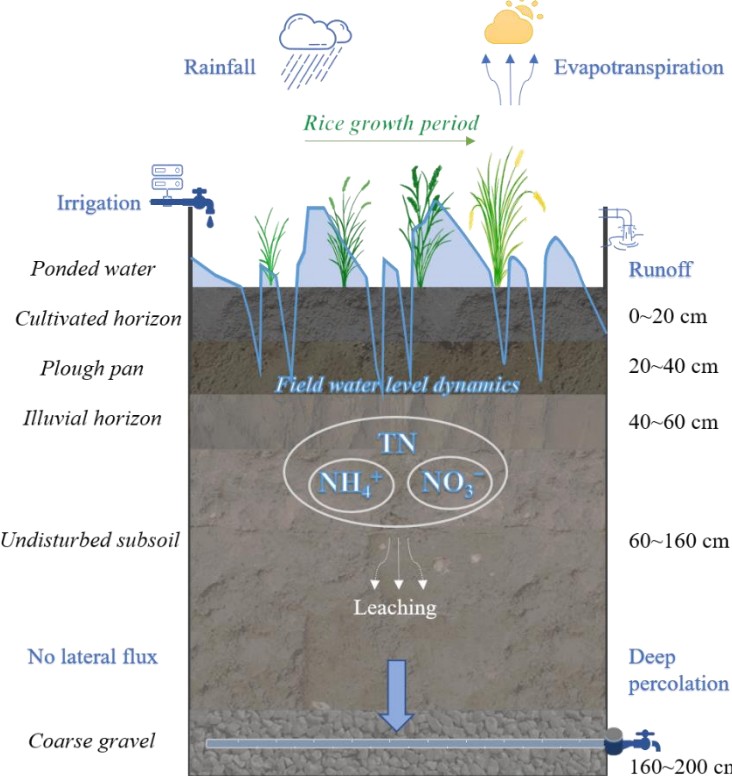

**Figure 1.** Schematic of the soil profile and water balance components of lysimeter experiments.

**Table 1.** Initial properties of the experimental soil at 0~40 cm depth.

| Property | 0~20 cm | 20~40 cm |
| --- | --- | --- |
| Sand % | 40.21 | 39.12 |
| Silt % | 38.22 | 39.16 |
| Clay % | 21.57 | 21.72 |
| Bulk density g cm$^{-3}$ | 1.38 | 1.41 |
| pH value | 6.94 | 6.97 |
| Total nitrogen g kg$^{-1}$ | 0.66 | 0.56 |
| Mineral nitrogen mg kg$^{-1}$ | 16.2 | 15.3 |
| Available phosphorus mg kg$^{-1}$ | 9.9 | 10.8 |
| Available potassium mg kg$^{-1}$ | 20.4 | 44.7 |
| Total organic matter % | 1.24 | 1.35 |

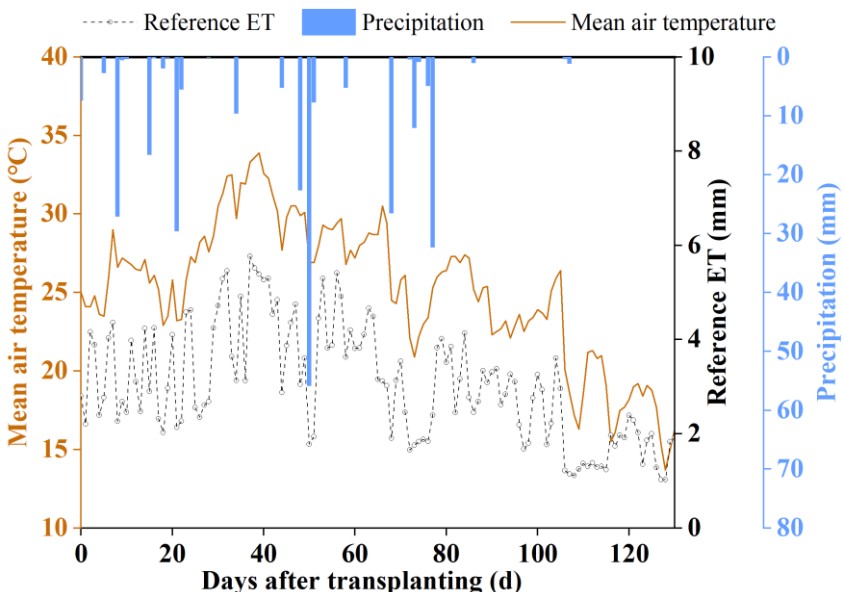

**Figure 2.** Variation of daily average air temperature, reference evapotranspiration and precipitation during the experiment in Jiangning, China. The reference ET refers to the reference crop evapotranspiration, which was calculated from the meteorological data of the experimental site.

### 2.2. Experimental Design

The experiment was conducted in a two-factor randomized complete block design with three duplicates. The W treatments consisted of two DP rates, 3 mm d$^{-1}$ and 5 mm d$^{-1}$, denoted as LP and HP, respectively. It means that when a surface water layer existed in the paddy field, the percolation rate would be controlled at 3 mm d$^{-1}$ or 5 mm d$^{-1}$ (converted to daily drainage of 15 L or 25 L), otherwise, there would be no drainage from the underground water outlet. Moreover, when the groundwater level dropped to the lower threshold, the plot was irrigated to 30 mm above the field surface. Both water treatments adopted CID practice, reducing surface runoff after rainfall to improve the efficiency of rainwater utilization [19]. As illustrated in Table 2, the timing and frequency of irrigation and drainage were based on the water level of each replicate [31]. Besides, the N treatments were N0~N4, representing 0, 60, 135, 210, and 285 kg N ha$^{-1}$, respectively. Specifically, N as urea was applied at the tiller initiation and spikelet-developing stages at a ratio of 1:1. For each treatment, 6000 kg ha$^{-1}$ wheat straw was returned to fields two weeks before transplanting, cut into 5 to 10 cm long and mixed into paddy soil. The rice cultivar was Nanjing 9108, which was recognized as high-yielding rice widely planted in local productions. It was transplanted on 21 June 2019 at a density of 20 cm × 15 cm, and harvested on 12 October 2019. Paddy fields were irrigated to 30 mm ponding in the returning green stage to promote the recovery of rice seedlings.

**Table 2.** Water management strategy under the controlled irrigation and drainage (CID) regime.

| Water Depth Criteria (mm) | | Returning Green | Tillering | Jointing-Booting | Heading-Flowering | Milky Ripening | Yellow Ripening |
|---|---|---|---|---|---|---|---|
| Irrigation | Lower threshold | 10 | −200 | −300 | −200 | −300 | Naturally drying |
| | Upper threshold | 30 | 30 | 30 | 30 | 30 | |
| Drainage | Upper threshold | 50 | 100 | 150 | 150 | 150 | |

*2.3. Sampling, Measurements and Calculation*

2.3.1. Irrigation and Drainage

Ponded water depths and groundwater table dynamics of each treatment were monitored each morning with a meter stick. Irrigation water was measured with an electromagnetic flowmeter installed at the pipe outlet of each lysimeter. When the ponded water depth exceeded the upper threshold due to rainfall, excess water was drained and measured with a flowmeter. The DP was drained from the underground water outlet at 9 o'clock every morning after observing the ponded water depth. The daily weather data were measured by an automated weather station established in the Jiangning Water-saving Park.

2.3.2. Nitrogen Concentrations in Field Water

The water samples were collected every 24 h within one week after each fertilization application, and then stored at $-20\ ^\circ$C for subsequent chemical analysis. Water samples were filtered by qualitative filter paper, and then the contents of N in ponded water or DP were measured by a UV–Vis spectrophotometer (UV-2800, Shimadzu, Kyoto, Japan). Specifically, $NH_4^+$-N, $NO_3^-$-N, and TN were quantified by Naismith's reagent UV spectrophotometry, hydrochloric acid acidification-UV spectrophotometry, and the alkaline potassium persulfate digestion UV spectrophotometric method, respectively. N leaching amount was the product of the drainage water volume and the determined N content.

2.3.3. Plant Analysis and N Uptake

Aboveground rice plants were sampled from each plot from tillering stage to maturity stage. Rice plant samples were weighed after oven-drying until a constant weight was reached. The tissue of dry matter was ground and then sieved with a 0.5 mm sieve. Subsequently, total N content was subjected to the Kjeldahl method, and digested with $H_2SO_4$-$H_2O_2$ at 260 $^\circ$C [32]. The N uptake of rice was the product of concentration and dry matter mass. At maturity, crops of each lysimeter were harvested separately to measure grain yield and yield components [32].

2.3.4. Water Productivity and NUE

Several indicators that consider uptake, assimilation and allocation of water and N during crop growth, such as *WP* (total water productivity, kg m$^{-3}$), *RE* (apparent recovery efficiency of N, %), *AE* (agronomic N use efficiency, kg kg$^{-1}$), *PFP* (partial factor productivity of applied N, kg kg$^{-1}$), and *PE* (physiological N use efficiency, kg kg$^{-1}$), were calculated according to the formulas in Djaman et al. [26] and Ding et al. [33]:

$$WP = \frac{Y}{P+I} \tag{1}$$

$$RE = \frac{U-U_0}{F_N} \tag{2}$$

$$AE = \frac{Y-Y_0}{F_N} \tag{3}$$

$$PFP = \frac{Y}{F_N} \tag{4}$$

$$PE = \frac{Y-Y_0}{U-U_0} \tag{5}$$

where $Y$ and $Y_0$ denoted the grain yield of N-fertilizer application treatments and N omission treatments, respectively (kg ha$^{-1}$); $P$ and $I$ denoted the precipitation and irrigation obtained by each treatment (m$^3$ ha$^{-1}$); $U$ and $U_0$ denoted the total N uptake of N-fertilizer application treatments and N omission treatments, respectively (kg ha$^{-1}$); and $F_N$ denoted the amount of N applied of each treatment (kg ha$^{-1}$).

*2.4. Statistical Analysis*

Data were analyzed by SPSS statistics 23 (SPSS Inc. IBM Corp., Armonk, NY, USA) and figures were drawn by Origin 2018 (OriginLab Corp., Northampton, MA, USA). Comparisons of means were tested with the least significant difference (LSD) method at $p < 0.05$. The main effects of water regimes (W), N application levels (N) and their interaction effects (W $\times$ N) were statistically analyzed at the 0.05 level.

## 3. Results

*3.1. Nitrogen Concentrations of Surface Water*

High N concentration in field water may diffuse through drainage and carry the risk of contaminating the surrounding water environment [34,35]. As illustrated in Figure 3a,b, the maximum NO$_3$$^-$-N concentration in ponded water was comprehensively lower than NH$_4$$^+$-N, and both increased with the larger amount of total N fertilizer input. Both LP and HP regimes showed a fluctuating trend of NO$_3$$^-$-N concentration, increasing within 3~4 days and decreasing subsequently. The NO$_3$$^-$-N concentration in ponded water at LP ranged from 0.81 to 7.54 mg L$^{-1}$, while it ranged from 1.21 to 7.39 mg L$^{-1}$ for HP. In particular, the difference in NO$_3$$^-$-N concentration was not significant ($p > 0.05$) between LP and HP. Additionally, the NH$_4$$^+$-N concentration in ponded water peaked on the second day after tillering fertilizer (TF) application, while NH$_4$$^+$-N peaked on the third day with spikelet-developing fertilizer (SF) and then declined gradually (Figure 3b). Moreover, the peak concentration of NH$_4$$^+$-N in ponded water after SF application was all lower than that of the same treatments after TF application. Without TF application (N0 treatments), the concentration of NH$_4$$^+$-N was 0.91 and 1.73 mg L$^{-1}$ under the LP and HP strategies, respectively. At HP, the maximum concentration of NH$_4$$^+$-N was 73.9 mg L$^{-1}$ with N4, 62.0% and 22.8% higher than those with N2 and N3 application levels, respectively. For N1 and N2 treatment, it took 2~4 days to reduce the NH$_4$$^+$-N concentration below 1.5 mg L$^{-1}$ to meet the surface water quality standards, while it took 5~7 days or more to meet the standards for N3 and N4 treatments. This means that paddy fields with higher N application levels need to pond for a longer time to avoid surface runoff loss and agricultural diffuse water pollution, especially after summer rainstorm events. The dynamics of the TN content in ponded water showed a similar pattern and magnitude with NH$_4$$^+$-N. After TF application, the TN content in ponded water peaked on the second day, and then decreased to stabilize within six to seven days (Figure 3c), and the ratio of NH$_4$$^+$-N to TN reached 77.1~88.7% due to the hydrolysis of urea. Reducing N fertilizer input may reduce the ammonium N and TN content in ponded water and the risk of loss with surface runoff from rice paddies for both TF and SF applications.

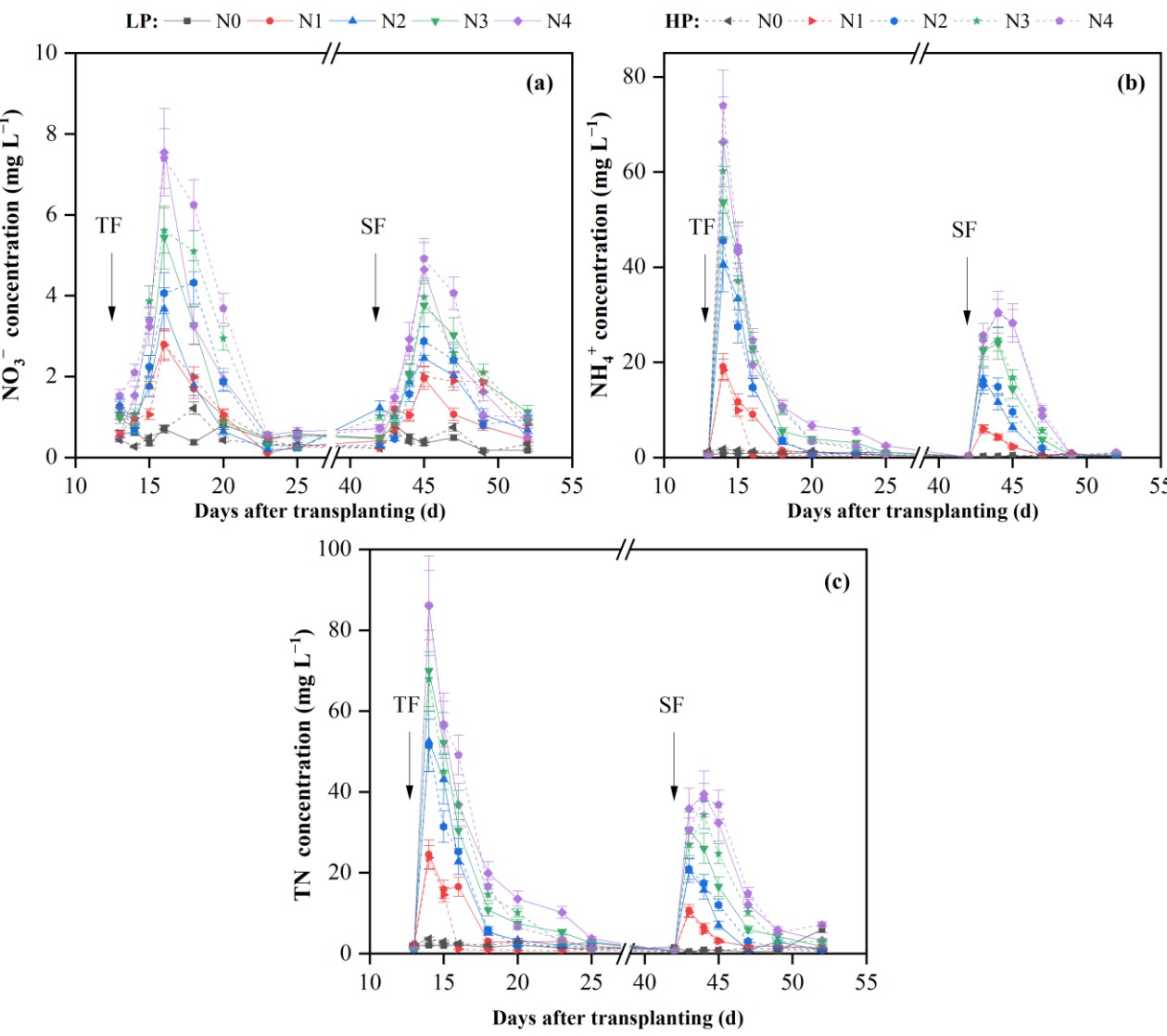

**Figure 3.** Variations of nitrate-N (NO$_3^-$-N, (**a**)), ammonium-N (NH$_4^+$-N, (**b**)) and total nitrogen (TN, (**c**)) concentrations in ponded water collected from different water and N treatments. The arrows denote the application of tillering (TF) and spikelet-developing fertilizer (SF), respectively. The application time of all treatments is the same. LP and HP, i.e., low percolation and high percolation, represent two deep percolation rates controlled by the lysimeters, 3 mm d$^{-1}$ and 5 mm d$^{-1}$, respectively. N0~N4 represent 0, 60, 135, 210 and 285 kg N ha$^{-1}$ application, respectively.

### 3.2. Nitrogen Loss via Deep Percolation

The DP under various water and N treatments is shown in Figure 4a. Percolation water volume at HP varied between 227.5 and 253.0 mm, where the DP of N0 treatment was significantly less than other N fertilizer treatments. Percolation water volume at LP varied between 144.0 and 162.0 mm, and no significant difference was observed at HP in different N levels. TN leaching losses reached 2.35~5.25 kg ha$^{-1}$ from controlled drainage paddies, which increased with increasing N inputs (Figure 4b). Analysis of variance showed that water and N management strategy had a significant main effect on TN leaching, and its interaction effect also reached a significant level ($p < 0.05$). It was observed that the N leaching loss of HP regimes was greater than that of LP regimes under the same N application rate. The TN leaching loss of HP treatment was 23.9% higher than that of LP on average. Moreover, N leaching losses accounted for about 1.6~6.8% of the N fertilizer inputs among different water and N treatments, indicating that N leaching losses during rice growth were not negligible. In addition, NO$_3^-$-N was the primary component of TN

losses with percolation, accounting for 86.5~90.8% and 83.6~89.3% of N losses under the LP and HP regimes, respectively. Thus, $NO_3^- $-N may be the main inducement to pollute the groundwater environment, and the application of high N fertilizer is particularly prone to aggravating the problem.

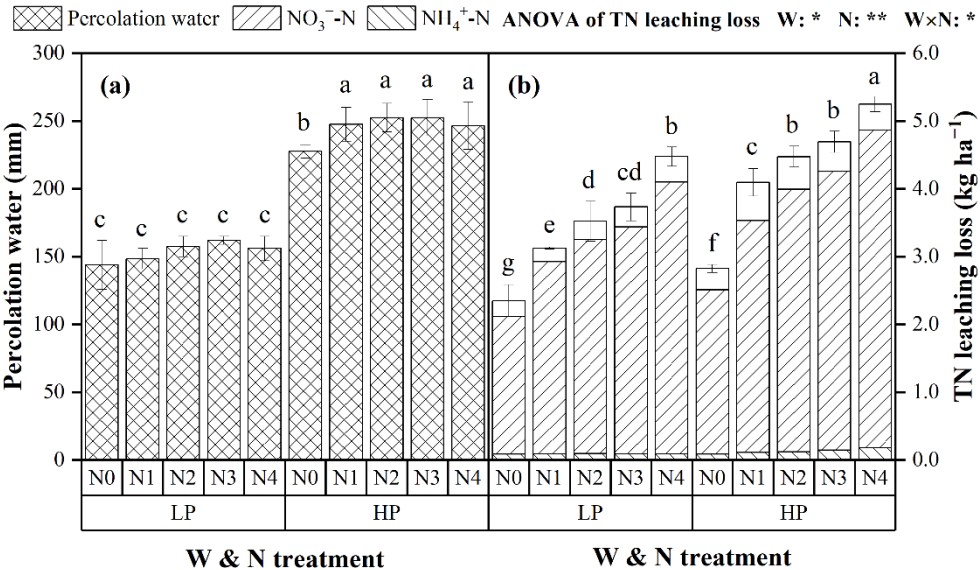

**Figure 4.** Percolation water (**a**) and total nitrogen (TN, (**b**)) leaching loss under various water and N treatments. Different letters next to means denote significant differences based on Least Significant Difference test (LSD test, $p < 0.05$). * and ** denote significant at 0.05 and 0.01 levels, respectively, while ns is non-significant ($p > 0.05$). LP and HP, i.e., low percolation and high percolation, represent two deep percolation rates controlled by the lysimeters, 3 mm $d^{-1}$ and 5 mm $d^{-1}$, respectively. N0~N4 represent 0, 60, 135, 210 and 285 kg N $ha^{-1}$ application with straw returning, respectively.

### 3.3. Grain Yield and Water Productivity

Rice grain yields were significantly increased between N0 and N2 rates under both different water regimes (Table 3). Higher yields occurred with the N2 to N4 rates, whose values ranged from 9.44 to 11.56 t $ha^{-1}$, and the maximum value of yield was observed at N3 rates in both DP levels; however, significant differences were not observed between the N2, N3 and N4 rates. Besides, from N3 to N4, the yield decreased by 0.4~3.2%. Applying more N-fertilizer only increased crop yield to a certain extent, and then crop demand for N reached saturation. Further over-fertilization had little positive or even negative effect on grain yield and the availability of fertilizer. The main effects of water and N showed significant effects on rice grain yield, while their interaction was insignificant. The panicles increased significantly with the increase of N application under both water regimes. In contrast, the filled spikelets fluctuated with an increasing N application rate and there was no significance observed in most water and N treatments. Based on the results of ANOVA, the interaction effect of W × N on the effective panicle number, spikelet per panicle, filled spikelets and 1000-grain weight was not significant. In addition, the N demonstrated the significant effects on the effective panicle number, spikelet per panicle and 1000-grain weight, while W significantly affected the effective panicle number, and spikelet per panicle.

The irrigation water volume during rice growth was 452.5 mm on average at LP, which was 82.3 % of that (549.9 mm) at HP (Figure 5). N4 treatment at HP significantly increased the consumption of irrigation water and reduced its *WP* correspondingly. In contrast to irrigation water, *WP* in LP was significantly greater than that in HP, and the responses showed a small difference in variations in both water regimes. With an increase in irrigation water input, *WP* ranged from 1.08 to 1.59 kg $m^{-3}$ and from 0.93 to 1.37 kg $m^{-3}$ under the LP and HP regimes, respectively. When the N application level increased from N0 to N2,

*WP* significantly increased. In addition, values in LP were higher than those in HP because of the large reduction in irrigation. N4 recorded the highest *WP* value at LP, while N3 recorded the highest *WP* value at HP. *WP* decreased by 10.2% from N3 to N4 at HP. Water or N management showed significant main effects on irrigation water and *WP*, but their interaction effects were not significant in both ANOVAs of irrigation water and *WP*.

**Table 3.** Grain yield and yield components under various water and N treatments.

| Water Treatment | N Treatment | Panicle ($\times 10^4$ ha$^{-1}$) | Spikelets (no. panicle$^{-1}$) | Filled Spikelets (%) | 1000-Grain Weight (g) | Grain Yield (t ha$^{-1}$) |
|---|---|---|---|---|---|---|
| LP | N0 | 238 d | 128 bc | 92.0 b | 24.7 a | 6.58 d |
|  | N1 | 263 d | 139 abc | 93.0 ab | 25.0 a | 8.10 c |
|  | N2 | 310 bc | 143 abc | 92.8 ab | 25.4 a | 9.88 ab |
|  | N3 | 329 ab | 152 a | 93.4 ab | 25.8 a | 11.56 a |
|  | N4 | 340 a | 147 ab | 92.6 ab | 25.0 a | 11.19 a |
| HP | N0 | 238 d | 123 c | 93.9 ab | 25.0 a | 6.44 d |
|  | N1 | 261 d | 137 abc | 92.6 ab | 25.3 a | 8.03 c |
|  | N2 | 287 c | 142 abc | 94.2 ab | 25.8 a | 9.44 ab |
|  | N3 | 303 bc | 149 ab | 95.5 a | 26.0 a | 11.18 ab |
|  | N4 | 329 ab | 146 ab | 93.0 ab | 25.2 a | 11.14 ab |
| ANOVA |  |  |  |  |  |  |
|  | W | * | ns | * | ns | * |
|  | N | * | ** | ns | * | ** |
|  | W $\times$ N | ns | ns | ns | ns | ns |

Note: Different letters next to means denote significant differences based on Least Significant Difference test (LSD test, $p < 0.05$). * and ** denote significant at 0.05 and 0.01 levels, respectively, while ns is non-significant ($p > 0.05$). LP and HP, i.e., low percolation and high percolation, represent two deep percolation rates controlled by the lysimeters, 3 mm d$^{-1}$ and 5 mm d$^{-1}$, respectively. N0~N4 represent 0, 60, 135, 210 and 285 kg N ha$^{-1}$ application, respectively.

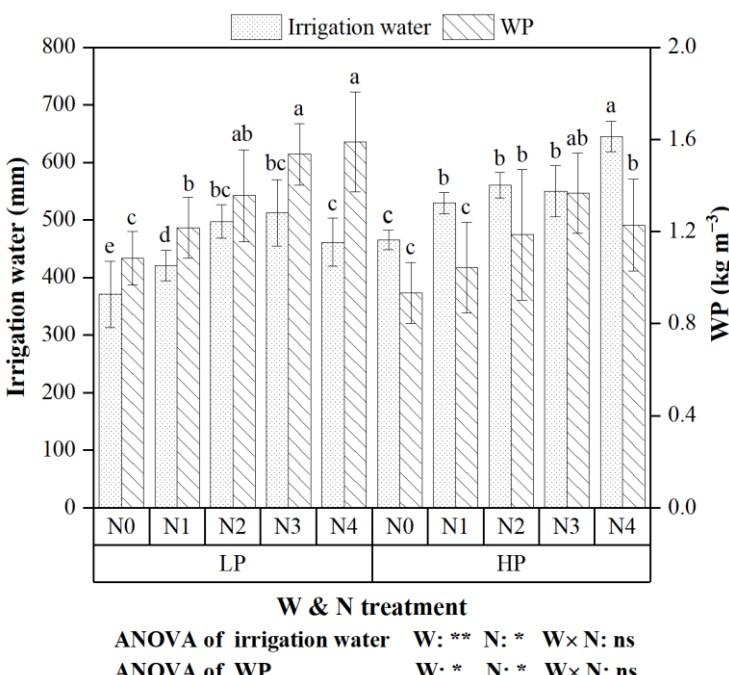

**Figure 5.** Irrigation water and total water productivity (*WP*) under various water and N treatments. Different letters next to means denote significant differences based on Least Significant Difference test (LSD test, $p < 0.05$). * and ** denote significant at 0.05 and 0.01 levels, respectively, while ns is non-significant ($p > 0.05$). LP and HP, i.e., low percolation and high percolation, represent two deep percolation rates controlled by the lysimeters, 3 mm d$^{-1}$ and 5 mm d$^{-1}$, respectively. N0~N4 represent 0, 60, 135, 210 and 285 kg N ha$^{-1}$ application, respectively.

*3.4. Nitrogen Uptake and Utilization*

The dynamics of total N accumulation in the above-ground organs of each treatment is shown in Figure 6a,c. The total N uptake increased with days after transplanting. The significant difference in total N accumulation between different N fertilizer application treatments was detected from the fourth sampling date (after the jointing period) under both LP and HP regimes. The N uptake of N3 and N4 was significantly greater than that of N0, N1 and N2 from the 24th day after transplanting at LP, whereas it began on the 35th day after transplanting at HP. Moreover, no significant differences were observed in the total N accumulation of N3 and N4 until the sixth sampling date (after the milky ripening period) at HP. From heading to the yellow ripen, N3 increased rice N uptake under the LP and HP regimes by 22.4% and 17.9%, respectively, while with the N4 level, it increased by 20.5% and 12.1% respectively. At maturity, the total N accumulation of N4 treatment in LP and N3 treatment in HP was the highest, 174.0 and 164.6 kg ha$^{-1}$, respectively (Figure 6b,d). When the N application level ranged from N0 to N3, the N uptake was significantly increased. It was observed that the total N accumulation of N4 treatment in HP was significantly lower than that of the N3 rate in HP. In addition, the total N accumulation in the panicles accounted for the largest proportion of total N accumulation in above-ground organs under the LP and HP regimes, ranging 71.2~81.0% and 73.4~81.2%, respectively, which decreased with increasing N level. According to the ANOVA at maturity, the effects of water and N were significant on N uptake, but the effect of W × N showed no significance.

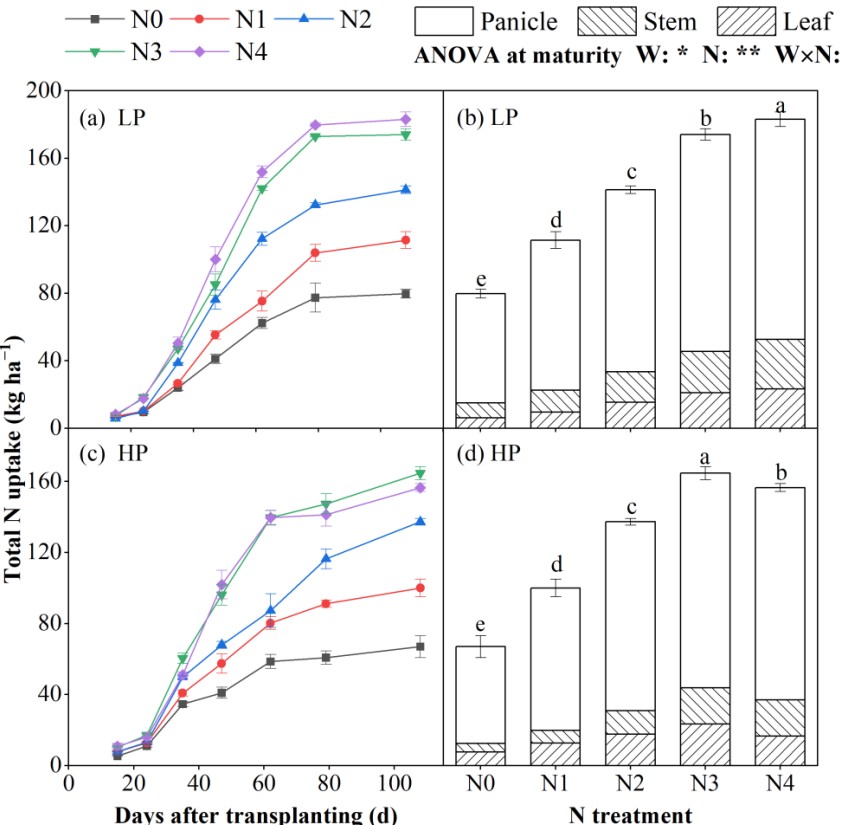

**Figure 6.** Dynamic of total nitrogen uptake (**left**, (**a,c**)) and TN accumulation in various organs of rice at maturity (**right**, (**b,d**)). Different letters next to means denote significant differences based on Least Significant Difference test (LSD test, *p* < 0.05). * and ** denote significant at 0.05 and 0.01 levels, respectively, while ns is non-significant (*p* > 0.05). LP and HP, i.e., low percolation and high percolation, represent two deep percolation rates controlled by the lysimeters, 3 mm d$^{-1}$ and 5 mm d$^{-1}$, respectively. N0~N4 represent 0, 60, 135, 210 and 285 kg N ha$^{-1}$ application, respectively.

The *AE* varied between 16.2 and 25.3 kg kg$^{-1}$ at LP and between 16.5 and 26.5 kg kg$^{-1}$ at HP (Table 4). *AE* was highest in N1 and lowest at N4 and showed no significant difference

between N2 and N3 under both water treatments. In addition, *PE* varied between 44.6 and 52.8 kg kg$^{-1}$ at LP and between 41.8 and 52.5 kg kg$^{-1}$ at HP. Changes in *PFP* were consistent with those in *RE* under both water treatments. In general, *AE*, *PFP* and *RE* reduced with the increase of N application from N1 to N4 under both water treatments, but the significance of the difference was inconsistent. For the LP and HP regimes, the *AE* of N4 treatment was 31.8% and 26.9% lower than that of N3, and the *PFP* of N4 treatment was 28.5% and 26.5% lower than that of N3, respectively. According to the results of ANOVA, the yield and *WP* of N3 and N4 were greater than those of N2, whereas the differences were non-significant, indicating that the marginal benefit of N-fertilizer input increment was decreasing. However, based on the indicators of *NUE*, N2 had higher *AE*, *PFP* and *RE* than N3 and N4; in particular, the differences in *NUE* indicators between N2 and N4 reached a significant level. For the LP and HP regimes, the highest N uptake and *WP* were obtained with N application levels of 285 kg ha$^{-1}$ and 210 kg ha$^{-1}$, respectively. In addition, either water or N showed significant effects on *PFP*, but no significant interaction effects of W × N were observed on *AE*, *PE*, *PFP* and *RE*.

**Table 4.** Agronomic N use efficiency (*AE*), physiological N use efficiency (*PE*), partial factor productivity of N (*PFP*), and apparent recovery efficiency of N (*RE*) under different water and N treatments.

| Water Treatment | N Treatment | *AE* (kg kg$^{-1}$) | *PE* (kg kg$^{-1}$) | *PFP* (kg kg$^{-1}$) | *RE* (%) |
|---|---|---|---|---|---|
| LP | N0 | - | - | - | - |
| | N1 | 25.3 ab | 48.0 ab | 135.06 a | 52.8 ab |
| | N2 | 24.4 b | 51.3 a | 73.2 b | 45.6 b |
| | N3 | 23.7 b | 52.8 a | 55.0 d | 44.9 b |
| | N4 | 16.2 d | 44.6 b | 39.3 e | 36.3 c |
| HP | N0 | - | - | - | - |
| | N1 | 26.5 a | 41.8 c | 133.8 a | 55.2 a |
| | N2 | 22.2 c | 42.6 b | 69.9 c | 52.1 ab |
| | N3 | 22.6 c | 48.5 ab | 53.2 d | 46.5 b |
| | N4 | 16.5 d | 52.5 a | 39.1 e | 31.4 c |
| ANOVA | | | | | |
| | W | ns | ns | * | ns |
| | N | * | ns | ** | * |
| | W × N | ns | ns | ns | ns |

Note: Different letters next to means denote significant differences based on Least Significant Difference test (LSD test, *p* < 0.05). * and ** denote significant at 0.05 and 0.01 levels, respectively, while ns is non-significant (*p* > 0.05). LP and HP, i.e., low percolation and high percolation, represent two deep percolation rates controlled by the lysimeters, 3 mm d$^{-1}$ and 5 mm d$^{-1}$, respectively. N0~N4 represent 0, 60, 135, 210 and 285 kg N ha$^{-1}$ application, respectively.

## 4. Discussion

Proper water and N management could have a positive impact on the improvement of yield [36,37]. The effects of water on grain yield (Table 3) and *WP* (Figure 5) were significant (*p* < 0.05), and N application significantly increased both irrigation amount and *WP* compared with N0 (Figure 5). Ren et al. [38] reported that the increased irrigation water input was due to increased water absorption, and the increased *WP* resulted from much higher rice yield with N application. More irrigation input enhanced rice yield but reduced the water use efficiency, which agreed with previous studies [39–41]. Meanwhile, yield components have been demonstrated to increase with the increase of N supply (Table 3), but the values of spikelets per panicle, filled spikelets and 1000-grain weight from N3 to N4 decreased slightly with the insignificance differences (*p* > 0.05). Ma et al. [42] suggested that higher N fertilizer treatment generally had higher yield due to a more effective N supply. Fertilization in excess of crop demand did not increase rice yield, and N surplus serves as a proxy for potential total N losses via ammonia volatilization, N leaching and denitrification [43]. High nitrogen loads from paddy fields pose a challenge to water environment protection in particular. It is therefore essential to optimize fertilizer inputs to reduce N losses for combating agricultural diffuse pollution.

As shown in Figure 5, the N0 treatment had the least amount of irrigation water under both water regimes ($p < 0.05$). There were differences in total water supply and water consumption among different N application treatments. Although their irrigation systems and schemes were consistent, their field water level dynamics and irrigation schedules were different. The higher N treatment may correspond to more vigorous growth demands [44]. Leaf and root development not only consumed nutrients, but also increased water evaporation and transpiration [45]. Therefore, more water supply was required for the high N treatments. Moreover, with increasing N application, the wet/dry cycles and irrigation times of the paddy increased under CID conditions [46]. The total subsurface drainage percolation was lower in the low N treatment than in the high N treatment (Figure 4) due to the lower irrigation frequency and total irrigation inputs. However, all plots were still managed strictly following the irrigation and drainage thresholds set by the CID system, which has been formulated according to rice growing stages and water requirements. In general, CID is a complex scheme that couples water-saving irrigation and controlled drainage technologies in rice paddies, and therefore its economic, environmental and technical characteristics deserve further research and validation.

The lowest TN detected in N0 in the absence of N fertilizer application may originate from the decomposition of straw and soil mineralization in alternating wet and dry rice fields. In ponded water, $NH_4^+$-N/TN reached a ratio of 77.1%, and the ratio increased with the N rate (Figure 3). The dominance of $NH_4^+$-N in ponded water may be due to the hydrolysis of urea and the suppression of nitrification by high water depth on the paddy surface after rainfall [47]. Another reason may be that part of the emitted ammonia may return to fields and surface water through deposition [48]. After fertilization, avoiding surface drainage may reduce nitrogen loss through surface runoff, and nutrients can be retained to improve *NUE*. Similar to the observation of Li et al. [49], lower percolation rates reduced groundwater recharge and associated N leaching loss. Additionally, the amount of $NO_3^-$-N leaching accounted for more than 83.6% of TN leaching loss, which was close to that of Xiao et al. [34], but was greater than other results found in rice paddies [50,51]. In general, the leaching of $NH_4^+$-N from the investigated soils was limited by negatively charged clay [52], while $NO_3^-$-N was not. Severe periodical wet/dry circles after the introduction of controlled drainage may also aggravate the N losses more than normal AWD [53]. Redox states of paddy soils at different depths also affected the process of nitrification and denitrification, thereby changing N transformation [54,55].

Straw was returned before transplanting in our experiment, and *AE* under different water and N treatments ranged from 16.2 to 26.5 kg kg$^{-1}$ (Table 4), which was higher than other results of the urea base fertilizer research [56,57]. This phenomenon was also summarized by a meta-analysis [33]. The improvement of N uptake under CID could be attributed to increases in root proliferation during the frequent drying cycle and the acquisition of nutrients from deeper soil layers [58], which may be another reason for the improvement of *NUE*. Research studies indicated that more gaseous N loss such as ammonia ($NH_3$) and nitrous oxide ($N_2O$) was observed in the continuous dry and wet alternation, but these losses did not affect N uptake significantly [59]. With more N-fertilizer input, the production of a large number of unproductive tillers was promoted in the vegetative stage, which wasted absorbed nutrients [41,60]. This reason also indirectly supported that *AE*, *PFP* and *RE* reduced with the increase of N application from N1 to N4 under both water treatments (Table 4). The root morphology involved in N uptake changes with water and nutrient status in the root zone and consequently influences plant growth and yield [61]. However, whether DP and N levels have main or interactive effects on root growth and their nutrient retention capacity merits further investigation.

## 5. Conclusions

Our study illustrated that $NH_4^+$-N was the major component in flooded water with controlled drainage, and its ratio to TN reached more than 77.1% and increased with the N application rate. $NO_3^-$-N was dominant in TN leaching losses, accounting for more than

83.6%, and the risk of groundwater contamination by nitrate leaching also increased with the increase of N application. High percolation treatments required more irrigation input, which not only had a negative effect on rice N uptake and yields, but also reduced *WP*. The proportion of N uptake in panicles was the largest in aboveground organs and decreased with increasing N application. The main effects of water or N on *NUE* and *WP* were significant, while their interaction effects were not significant. Further research is needed to investigate the processes of rice root growth and nutrient retention and understand the mechanism of N transport and transformation in the paddy environment by designing more controlled drainage scenarios under different hydrological years and soil conditions.

**Author Contributions:** Conceptualization, S.Y. and J.D.; Funding acquisition, S.Y.; Methodology, K.C. and P.H.; Formal analysis, K.C. and T.M.; Investigation, K.C., Y.D. and G.Z.; Writing—original draft preparation, K.C.; Writing—review, editing, and supervision, K.C. and T.M. All authors have read and agreed to the published version of the manuscript.

**Funding:** This research was funded by the National Natural Science Foundation of China (Grant No. 51879074 and 52109051), the Postgraduate Research & Practice Innovation Program of Jiangsu Province (Grant No. KYCX21_0540), and with financial support from the Natural Science Foundation of Jiangsu Province (Grant No. BK20200513).

**Institutional Review Board Statement:** Not applicable.

**Informed Consent Statement:** Not applicable.

**Data Availability Statement:** All data are provided as tables and figures.

**Acknowledgments:** We are grateful for the help of Qiong Wang, Mengting Zhang, Zixin Liu and Yu Wang in the implementation of the experiment.

**Conflicts of Interest:** The authors declare no conflict of interest.

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
