# Peer review of "Effects of Water and Nitrogen Management on Water Productivity, Nitrogen Use Efficiency and Leaching Loss in Rice Paddies"

_water, doi:10.3390/w14101596_

Round 1

Reviewer 1 Report

The manuscript „Effects of Water and Nitrogen Management on Water Productivity, Nitrogen Use Efficiency and Leaching Loss in Rice Paddies” covers important aspects of sustainable agronomic production. In general it is methodologically sound and a good read. However, there still several issues which need to be addressed before it can be considered for publication.

1.    The description of the tested irrigation system CID is far too short for an international audience, it should be explained in more detail.
2.    In how far does the lysimeter system/soil represent typical Chinese paddy soils. What about the plough pan and its importance for leaching losses? How could variability in soil conditions affect observed effects?
3.    Technical terms partly not clear. What does deep percolation mean? Does it imply leaching losses below the root zone?
4.    Seasonal effects: the study covers only one growing season which is probably too short to derive general recommendations. What efforts are still needed to obtain a more robust result, including effects of weather and soil conditions.
5.    Optimal N rates cannot be directly derived from specific treatment levels but from yield response curves. It is strongly recommended to make a more appropriate analysis for this purpose.
6.    Statistical analysis: as the experimental design was a split plot design it should also statistically analysed by a split-plot ANOVA. Statistical analysis needs to be reworked.
7.    Other emission processes, e.g. NH3 emissions, should be discussed concerning N use efficiency.
8.    Some graphs are hard to read Fig. 2, Fig 5 a,c

Authors are strongly encouraged to include the requested information, analyses and discussion.

Points in detail
L 10: are critical for sustainable….
L 12 delete ‘the’ in front of controlled
L 14 results demonstrated that…
L 23 optimal N rate should be derived in a different way, not by analysis of variance
L 25 reduction compared to which reference?
L 35 50% of what, used water or total fresh water?
L 78: this sentence is not clear, why should CID increase leaching, when rainfall water is stored in place of irrigation water?
L 91ff See general criticism on representativeness of lysimeter soils, the setup of the lysimeter design should be motivated here.
CID system should also be explained in some more detail here
L 180 High N concentration in field water…
L 220 interaction effects should also be presented here
L 244 excessive, please rephrase, eventually ‘fertilization in excess of crop demand’
L 319 do not agree to this conclusion: a rate of 135 kg N/ha is lower than crop uptake which indicates soil mining. Soil mining for nutrients is considered as an unsustainable practice (see NUE concept of European Nitrogen Expert Panel). Crop uptake should not exceed 90% of fertilizer N applied.
L 336 enhanced rice yield (delete 'could')
L 341 ff that is no new information and corresponds to textbook knowledge 
L 358ff see my comment on representativeness of soil conditions, sentence unclear
L 366 the main cause for dominance of NH4+ is probably that high water levels supress nitrification
L 371 off sentence, please rephrase
L 388 ff please do not repeat results in the conclusions, only main findings and further research demand derived from those.
L 400 ff this sentence is odd and tautological, excessive N application is always uneconomic compared to optimal N rate. Please rephrase sentence.
L 405 do not agree to this conclusion. The chosen analysis was not appropriate to derive optimal N rate (see above comments)
L 406 ff this statement comes ‘out of the blue’, there was no mentioning of this aspect throughout the manuscript.

Reviewer 2 Report

Dear authors,

congratulation on demanding fieldwork, quality results, and interesting manuscript. I greatly enjoyed reading your study results, the quality of the presentation is high and I have only a minor suggestion about the conclusion. Kindly see my suggestion: 

The conclusion section should be shorter than the current one, more concise, and with the most important results that would attract readers. Try to draw the most important conclusions and briefly suggest which results are most important for the practice of irrigation and N fertilization.

Wish you good luck in future scientific work,

Best regards

This section should be shorter than the current one, more concise, with the most important results that would attract readers. Try to draw the most important conclusions and briefly suggest which results are most important for the practice of irrigation and N fertilization.

Round 2

Reviewer 1 Report

Dear authors,

the manuscript was strongly improved by the revision. However, there are still some points which require consideration before the manuscript can be accepted for publication.

It is still not clear enough which novelty is connected to the presented research. Water and nitrogen interactions are not new with respect to paddy rice production. So you should focus on your specific irrigation scheme CID and what was the learning with respect to it in your research. This should be given in the abstract, the discussion and the conclusions.

In addition, it would be good to further improve the readability of the line graphs, if possible, and to check for still existing language issues.

Some detailed points:

L 29 acceptable no appropriate term, please specify and replace by different word

L 30 the same for ‘reasonable’, quite colloquial wording, what does reasonable mean?

L 93 increased ammonia volatilization

L 137ninitial properties where, plough layer?

Was water supply to all treatments the same? How can reduced leaching for 0N explained

L 358 not yield, rather N uptake

Stronger highlighting of novelty of research

L 391 ff odd sentence please rephrase

L 392 based on rice and water supply prices the economy of the treatments could also be calculated

L 395 ff this sentence is odd and a verb is missing in the second half

Author Response

This manuscript is a resubmission of an earlier submission. The following is a list of the peer review reports and author responses from that submission.